# Novel Approaches in Molecular Characterization of Classical Hodgkin Lymphoma

**DOI:** 10.3390/cancers14133222

**Published:** 2022-06-30

**Authors:** Diede A. G. van Bladel, Wendy B. C. Stevens, Michiel van den Brand, Leonie I. Kroeze, Patricia J. T. A. Groenen, J. Han J. M. van Krieken, Konnie M. Hebeda, Blanca Scheijen

**Affiliations:** 1Radboud University Medical Center, Department of Pathology, 6525 GA Nijmegen, The Netherlands; diede.vanbladel@radboudumc.nl (D.A.G.v.B.); mvandenbrand@rijnstate.nl (M.v.d.B.); leonie.kroeze@radboudumc.nl (L.I.K.); patricia.groenen@radboudumc.nl (P.J.T.A.G.); han.vankrieken@radboudumc.nl (J.H.J.M.v.K.); konnie.hebeda@radboudumc.nl (K.M.H.); 2Radboud Institute for Molecular Life Sciences, 6525 GA Nijmegen, The Netherlands; 3Radboud University Medical Center, Department of Hematology, 6525 GA Nijmegen, The Netherlands; wendy.stevens@radboudumc.nl; 4Pathology-DNA, Rijnstate Hospital, 6815 AD Arnhem, The Netherlands

**Keywords:** classical Hodgkin lymphoma, clonality assessment, next-generation sequencing, mutation analysis, cell-free DNA

## Abstract

**Simple Summary:**

The unique tumor composition of classical Hodgkin lymphoma (cHL), with only a small fraction of malignant Hodgkin and Reed–Sternberg cells within the tumor tissue, has created many challenges to characterize the genetic alterations that drive this lymphoid malignancy. Major advances in sequencing technologies and detailed analysis of circulating tumor DNA in blood samples of patients have provided important contributions to enhance our understanding of the pathogenesis of cHL. In this review, we provide an overview of the recent advances in genotyping the clonal and mutational landscape of cHL. In addition, we discuss different next-generation sequencing applications to characterize tumor tissue and cell-free DNA, which are now available to improve the diagnosis of cHL, and to monitor therapeutic response or disease progression during treatment and follow up of cHL patients.

**Abstract:**

Classical Hodgkin lymphoma (cHL) represents a B-cell lymphoproliferative disease characterized by clonal immunoglobulin gene rearrangements and recurrent genomic aberrations in the Hodgkin Reed–Sternberg cells in a reactive inflammatory background. Several methods are available for the molecular analysis of cHL on both tissue and cell-free DNA isolated from blood, which can provide detailed information regarding the clonal composition and genetic alterations that drive lymphoma pathogenesis. Clonality testing involving the detection of immunoglobulin and T cell receptor gene rearrangements, together with mutation analysis, represent valuable tools for cHL diagnostics, especially for patients with an atypical histological or clinical presentation reminiscent of a reactive lesion or another lymphoma subtype. In addition, clonality assessment may establish the clonal relationship of composite or subsequent lymphoma presentations within one patient. During the last few decades, more insight has been obtained on the molecular mechanisms that drive cHL development, including recurrently affected signaling pathways (e.g., NF-κB and JAK/STAT) and immune evasion. We provide an overview of the different approaches to characterize the molecular composition of cHL, and the implementation of these next-generation sequencing-based techniques in research and diagnostic settings.

## 1. Introduction

Classical Hodgkin lymphoma (cHL) is one of the most common lymphoma types in the Western world, with an age-adjusted incidence rate of about 3 cases per 100,000 people [1]. Most cHL patients are diagnosed around young adulthood, followed by a second peak at the age of 60 years and older. There are four different histological subtypes defined for cHL, of which nodular sclerosis (NSCHL) and mixed cellularity (MCCHL) are the most common [2].

The cellular composition and morphology of cHL are unique among the different B-cell lymphomas. cHL consists of a minor neoplastic B-cell component (usually <1–5%) represented by enlarged mononucleated Hodgkin cells and multinucleated Reed–Sternberg cells, referred to as Hodgkin and Reed–Sternberg (HRS) cells, which are surrounded by a variable mixture of inflammatory cells, including reactive B- and T-cells, histiocytes and granulocytes [3,4,5]. The diagnosis for most patients is based on clinical presentation and histopathology, including the detection of CD30-positive HRS cells. Occasionally, cHL is difficult to distinguish from reactive lymphoproliferations or B- or T-cell lymphomas with HRS-like cells [6]. In such cases, molecular assays, including the detection of clonal immunoglobulin (IG) and T-cell receptor (TR) gene rearrangements or gene mutation analysis, can be essential for the correct diagnosis and treatment. Besides de novo cHL, this specific lymphoma subtype may occasionally develop as a transformation from low-grade B-cell lymphoma [7,8]. Although cHL is highly curable with multiagent chemotherapy and modern radiation techniques, up to 30% of patients with advanced stage cHL have progressive disease or develop a relapse [2,9,10]. Clonality testing and mutational profiling can be valuable techniques to assess the clonal relationship between the primary tumor and the clinical recurrence or a second lymphoid malignancy.

Because of the small fraction of neoplastic HRS cells in cHL, conventional clonality assays and mutation analysis have often failed to detect the molecular profiles of the HRS cells. With specialized techniques to isolate or enrich HRS cells, mutation and clonality analyses appeared technically more feasible, but these approaches are time- and labor-intensive and therefore not suitable for clinical practice. The major advances of next-generation sequencing (NGS) technologies and analysis of cell-free DNA (cfDNA) in the blood of lymphoma patients in the last decade have made important contributions to improve our understanding of the molecular pathogenesis of cHL [11,12,13]. Despite the relatively low number of HRS cells in cHL tumor tissues, there is a relatively high concentration of tumor-derived cfDNA, also known as circulating tumor DNA (ctDNA), present in the peripheral blood of most reported cHL patients, with potential for improved diagnostics and monitoring of cHL patients during treatment [14]. Here, we provide an overview of the different NGS-based molecular approaches that are available to characterize cHL in both tumor tissue and cfDNA.

## 2. Clonality Assessment in Classical Hodgkin Lymphoma Tissue

During early B- and T-lymphoid development, the different germline variable (V), diversity (D) and joining (J) genes of the IG and TR genes are recombined, a process called V(D)J recombination (Figure 1). This recombination process results in a unique antigen-receptor molecule on the surface of the mature lymphocytes. During further maturation of B cells within the germinal centers, the B lymphocytes extend their IG repertoire for antigen recognition by somatic hypermutation (SHM) [15,16]. The unique clonal IG/TR gene rearrangements, due to the shared cell of origin of the malignant lymphoid cells within one tumor, make the detection of these tumor-specific V(D)J rearrangements (i.e., clonotypes) a highly valuable molecular marker for clonality testing of lymphoproliferative diseases. Although cHL is characterized by the lack of a B-cell receptor on the cell surface, which is in contrast to most other B-cell lymphomas [17], genomic IG gene rearrangements have occurred during the premalignant B-cell development stage. These IG gene rearrangements can therefore be detected as a molecular fingerprint for an individual cHL tumor, and may involve both functional and non-functional IG gene rearrangements [17,18]. The low number of HRS cells in a background of many reactive immune cells, including B cells, and the high level of SHM potentially hampering efficient primer binding, makes detection of clonal IG rearrangements challenging in cHL whole tumor tissue specimens. Furthermore, the presence of reactive B cells within the tumor microenvironment (TME) may further complicate the analysis of clonality testing in cHL. These non-malignant B cells may undergo clonal expansion, which eventually results in minor clones.

Initially, clonality detection was performed using Southern blot analysis, with a large variety in the IG clone detection rate for cHL cases, ranging from 0% to 87% [19,20,21,22,23,24,25,26]. The introduction of more sensitive PCR-based techniques improved clonality detection for this type of lymphoma [27,28,29] (Table 1, upper panel). In the early 2000s, clonality detection was further optimized by the introduction of standardized multiplex PCR protocols as developed by the BIOMED-2 Concerted Action, nowadays called the EuroClonality consortium [30]. This BIOMED-2/EuroClonality assay in combination with GeneScan fragment size analysis yielded a mean clonality detection rate of 57% (range 26–79%) in cHL (Table 1, upper panel) [31,32,33,34,35,36,37,38,39]. The improvements in clonality testing based on the BIOMED-2/EuroClonality assay were achieved by the complementarity of the assays using multiple targets in both IGH and IGK loci [40,41]. Other groups introduced an additional semi-nested PCR protocol [35]. Other approaches that have been employed for the detection of IG gene rearrangements in cHL involved the enrichment or isolation of single HRS cells, which obviously increased the sensitivity and specificity of clonality detection (Table 1, upper panel) [17,18,26,42,43,44,45,46,47,48,49]. However, not every study was able to detect IG gene rearrangements in these cases, which could relate to the lack of HRS cell nuclei in thin tissue sections used for isolation, or mismatches of consensus primers due to SHM.

An important recent innovation for clonality assessment involved the introduction of NGS-based clonality testing on tissue biopsies by the EuroClonality-NGS Working Group [55,56]. This approach clearly improved clonality assessment in cHL as compared to the BIOMED-2/EuroClonality assay [51]. Other NGS-based assays, either amplicon-based (e.g., ClonoSEQ technology [50]) or hybrid capture-based [52], have also been used for clonality detection in cHL. These approaches have resulted in higher clone detection rates (range 38–89%) compared to conventional clonality testing, and included analysis in both cHL tissue DNA and cfDNA (Table 1, upper panel) [50,51,52]. A clear advantage of NGS-based clonality testing is the availability of the exact clonotype sequence. This allows for the specific detection of minor clones in a polyclonal background, as is the case in cHL. Several studies have also investigated the presence of clonal TR gene rearrangements in cHL (Table 1, lower panel). Although cHL is considered a B-cell lymphoma, the detection of clonal TR gene rearrangements in isolated HRS cells has suggested a T-cell origin in some cases of cHL [42,47,53,54,57,58]. TR clonality testing can also assist in the differential diagnosis of cHL with Hodgkin mimickers, such as angioimmunoblastic T-cell lymphoma (AITL) with a Hodgkin-like B-cell component or the follicular variant of peripheral T-cell lymphoma [36,59,60].

Finally, clonality assessment is highly informative to establish the clonal relationship of cHL recurrences, which has been investigated in only one study with a significant cohort size [61]. For 20 cHL patients with cHL recurrences, laser microdissection (LMD) was performed followed by Sanger sequencing of isolated HRS cells to compare tumor-specific clonotypes in the primary and subsequent cHL presentation. This study showed that in 8 out of 20 cHL patients (40%), the primary diagnosis and recurrence represented clonally unrelated lymphomas, suggesting the occurrence of second primary lymphoma in a significant number of cHL patients. Obviously, additional studies in larger cHL patient cohorts are required to confirm this observation.

## 3. New Developments in Molecular Testing of Classical Hodgkin Lymphoma with cfDNA

Currently, there is emerging interest in cfDNA analysis as a tool for non-invasive tumor detection and monitoring in clinical practice. Although cfDNA is commonly present in healthy subjects due to apoptosis of hematopoietic cells, studies show that cfDNA of cancer patients harbors tumor-derived fragmented DNA, known as circulating tumor DNA (ctDNA), which may provide molecular information related to the tumor [62]. The high sensitivity of sequencing techniques and droplet digital PCR (ddPCR) have enabled the use of ctDNA for early tumor detection, molecular profiling and monitoring of therapy response or disease progression [14,52,63,64,65,66,67]. Furthermore, the introduction of unique molecular identifiers (UMIs) for error suppression during next-generation sequencing [68,69] has resulted in even more sensitive tests for mutation detection in diagnostic settings. Since cfDNA has a short half-life of about 15 min to a few hours in the bloodstream, analysis of this type of DNA reflects the mutation status of the tumor at time of sampling, making it highly suitable for disease monitoring during treatment [70]. The analysis of cfDNA may also be of added value when tissue sampling is difficult due to localization of the tumor, or for better characterization of a heterogeneous tumor [70]. Of note, ctDNA isolation from plasma after blood sampling requires a special workflow of either EDTA tubes (processing time < 4 h) or nucleic acid preservation tubes (Roche or Streck cfDNA BCT; processing time < 24 h). A short processing time is necessary to avoid degradation of leukocytes and subsequent contamination of the ctDNA fraction.

A substantial part of cfDNA that is present in blood is derived from hematopoietic cells and may harbor non-tumor-derived mutations as a result of clonal hematopoiesis [71]. Clonal hematopoiesis is a normal process of aging and involves the accumulation of somatic mutations in hematopoietic stem cells, leading to clonal expansion of such mutations in blood cells [72]. The genes most frequently affected in clonal hematopoiesis are DNMT3A, TET2, ASXL1, TP53, JAK2, SF3B1, SRSF2 and PPM1D [73,74,75,76]. Although there is only limited overlap with cHL-associated genes, the presence of such variants in cfDNA of lymphoma patients may complicate ctDNA analysis, unless paired deep sequencing of plasma cfDNA and DNA from white blood cells has been performed [77].

One of the first studies providing evidence that genetic features of HRS cells can be detected in cfDNA involved non-invasive prenatal testing in an asymptomatic pregnant woman with an abnormal genomic profile who subsequently was diagnosed with cHL [78]. The possibility to detect genetic features in cfDNA was confirmed in nine additional patients with newly diagnosed cHL. Subsequent studies have confirmed the potential added value of analyzing cfDNA for cHL diagnostics and monitoring [52,63,64,65,66,67]. Using targeted sequencing, tumor-derived somatic mutations in cfDNA were identified with high detection rates, which were validated by the analysis of paired tumor tissue biopsies. In a recent study, gene-specific mutations were detected with ddPCR in cfDNA samples, demonstrating that this approach is also feasible in cHL diagnostics [79]. Taken together, these studies have shown that genotyping of cfDNA in cHL allows for at least equal performance of mutation detection as in whole tissue. This might be related to the high turnover rate of HRS cells, resulting in relatively larger amounts of ctDNA, despite the low HRS cell number in tumor tissues [14].

Potentially clinically relevant applications of cfDNA involve the detection of minimal residual disease (MRD) to monitor therapy response, as has been applied in different lymphoma types [80,81], including cHL [50,82], and for prognostic stratification [14,52,82,83,84,85]. This requires sensitive NGS-based assays, such as ClonoSEQ (formerly known as LymphoSIGHT) [81,86], CAPP-Seq [87], PhasED-Seq [88], or the EuroClonality-NGS DNA capture (EuroClonality-NDC) assay [89]. Despite the promising results obtained in different laboratories that have performed mutation and clonality analysis in cfDNA of cHL patients, there are still several challenges regarding the sensitivity and reproducibility of these assays. Large prospective clinical studies are required before cfDNA testing can be implemented in routine diagnostics and monitoring of cHL.

## 4. Molecular Pathogenesis of Classical Hodgkin Lymphoma

Accumulating genetic alterations within the HRS cells and infection by Epstein–Barr virus (EBV) play an important role in the pathogenesis of cHL [11,13,90]. Given the scarcity of the malignant HRS cells in cHL tissue, major advances to unravel the molecular pathogenesis of cHL have been achieved by whole exome sequencing (WES) on enriched HRS cells [91,92,93]. This confirmed the high genomic instability of HRS cells that was known from cytogenetic studies [12,94,95,96], which includes somatic mutations, translocations involving, among others, IG loci with various translocation partners [97], and recurrent copy number alterations (CNAs), including gains of chromosomal arms 2p, 9p, 16p, and 17q and losses of 13q, 6q, and 11q [95,98,99,100]. In fact, 9p24.1 gains are detected in almost all cHL cases and represent a hallmark of cHL. The genes encoding JAK2, as well as PD-1 ligands PD-L1 and PD-L2, are located on this locus, which all play an important role in cHL pathogenesis [101]. More recent studies were also able to detect mutations in genomic DNA of cHL whole tissue specimens, using targeted deep sequencing [79]. These approaches have been complemented by hybrid capture-based sequencing of ctDNA, which has allowed the detection of both tumor-specific gene mutations [52,63,64,65,66,67], and CNAs in cHL [102].

### 4.1. Mutational Landscape of Classical Hodgkin Lymphoma

Detailed molecular analyses of HRS cells and ctDNA in cHL patients have revealed the presence of recurrent gene mutations and genomic alterations that are involved in signaling pathways promoting cell proliferation and survival, as well as immune evasion (Figure 2 and Table 2) [14,52,64,66,67,79,91,92,93]. The most frequently affected pathway in cHL represents the JAK/STAT signaling pathway, which plays a central role in promoting cell proliferation and survival. JAK/STAT signaling is constitutively activated in HRS cells, as a result of inactivating mutations in its negative regulators, such as SOCS1 and PTPN1, activating mutations in STAT6 and overexpression of JAK2, due to focal gene amplifications or copy number gains of 9p24. HRS cell proliferation and survival are also regulated by the NF-κB pathway, which is constitutively activated by genetic alterations targeting TNFAIP3, NFKBIA, NFKBIE, IKBKB, and REL. In addition, HRS cells escape antitumor immune responses via inhibition of PD-1-expressing immune cells, such as cytotoxic T cells, due to overexpression of the PD-1 ligands PD-L1 and PD-L2. Other factors that induce immune evasion of HRS cells include inactivating alterations of B2M and CIITA, encoding components or regulators of MHC class I and II expression, respectively. Inactivating B2M mutations have been related to the NSCHL subtype, and a lack of B2M correlates with better clinical outcomes [91]. STAT6 and TNFAIP3 are more frequently mutated in NSCHL, and STAT6 mutations are more often observed in younger patients [67]. Others have also observed a correlation between ITPKB and B2M mutations and more disseminated disease [66]. However, additional studies are required to confirm the prognostic value of specific gene alterations and associations with certain cHL subtypes.

Other pathways and gene functions that are perturbed in cHL include the PI3K/AKT pathway, chromatin remodeling, nuclear–cytoplasmic transport and the p53 pathway. Activation of the PI3K/AKT pathway involves mutations in ITPKB and PIK3CA, which, together with inactivating GNA13 mutations [103,104,105], promote lymphoma cell survival. Gene mutations affecting proteins involved in chromatin remodeling are frequently observed in different B-cell malignancies. In cHL, the tumor suppressor gene ARID1A is among the most frequently mutated genes, and a component of the SWI/SNF chromatin remodeling complex. Recently, it was shown that ARID1A deficiency correlates with a higher mutation load and improved sensitivity to immune checkpoint blockade using anti-PD-L1 treatment in a preclinical ovarian cancer model [106]. It remains to be determined whether ARID1A mutations show predictive value in cHL immune checkpoint blockade. Furthermore, about 20% of analyzed cHL patients show mutations in XPO1, of which the majority shows the E571K hotspot mutation [52,64,65,67,92,93]. This gene is involved in nuclear–cytoplasmic transport and its alteration results in disturbed homeostasis of HRS cells. Interestingly, detectable XPO1 mutations at the end of treatment seem to result in shorter progression-free survival compared to patients with undetectable XPO1 mutations after treatment, and could potentially be used as a predictive marker [65].

The importance of the p53 pathway in cHL pathogenesis has remained controversial. In some studies, TP53 mutations were found to be rare (<10%) [52,92,107], while others reported higher frequencies of TP53 mutations (~25%) [79,108,109]. It is very plausible that EBV status may act as a confounding factor, since TP53 mutations are mainly detected in EBV-negative cases [108]. Mutations in TP53 can stabilize p53 protein due to conformational changes, which is visible by increased expression of p53 protein by immunohistochemistry, as is frequently observed in the nuclei of HRS cells [107,108,110]. Since mutated p53 protein can promote the survival of genomically instable cells, such as HRS cells, it is likely that reduced p53 function plays a role in the pathogenesis of cHL.

### 4.2. The Role of Epstein–Barr Virus Infection in Classical Hodgkin Lymphoma Pathogenesis

In 20–40% of the cHL cases in the Western world and more than 90% of pediatric cHL cases in developing countries, the lymphoma is EBV-positive, and there is compelling evidence that EBV serves as a tumor-initiating factor in cHL [5,111,112,113]. There are various EBV infection patterns known for lymphoid cells, each with a different range of EBV latent antigen expression. EBV-infected HRS cells in cHL display the latency type II program with expression of EBV latent genes LMP1, LMP2A, LMP2B and EBNA1 together with EBER non-coding RNAs and BART miRNAs, which all play a role in the pathogenesis of cHL (Figure 3) [113,114]. In most B cells, expression of the BCR serves as a survival signal. Since HRS cells lack expression of a functional BCR, LMP1 and LMP2A rescue HRS cells from apoptosis. Transmembrane protein LMP1 shows functional similarities with constitutive active CD40 signaling, while LMP2A functions as a BCR mimic with the ability to activate the RAS/MAPK pathway [90,113,115,116]. Together, these EBV proteins mediate activation of downstream signaling pathways (e.g., NF-κB, JAK/STAT and PI3K/AKT), and modulate the expression of cellular genes regulating B-cell proliferation, survival, lineage commitment/differentiation and immune escape [114,117]. The EBV latent protein EBNA1 is a key viral replication factor and important for maintaining the viral genome during cell division. In addition, it is capable of acting as a transcription factor, thereby regulating gene expression that promotes cell survival and contributing to immune evasion of EBV-infected HRS cells [90,118]. EBV-encoded small RNAs, or EBERs, inhibit apoptosis and induce the transcription of cytokines, resulting in HRS cell survival [114,119,120]. The exact roles of BART miRNAs in cHL pathogenesis are just starting to be explored. BART miRNAs contribute to HRS cell survival and reduction in tumor elimination by the immune system [121]. The pathogenic role of EBV infection in cHL development is further supported by the observation that EBV-negative cHL cases show a significantly higher mutational burden and more chromosomal abnormalities, compared to EBV-positive cases [92,93,122,123,124]. This underscores the finding that these viral proteins substitute the requirement for specific oncogene and tumor suppressor gene alterations in cHL pathogenesis. Indeed, TNFAIP3 and TP53 mutations and aberrant activation of receptor tyrosine kinases occur predominantly in EBV-negative cases [111,125,126].

### 4.3. T-Cell Immune Microenvironment in Classical Hodgkin Lymphoma

The tumor microenvironment (TME) of cHL is unique, as it contains a large admixture of inflammatory immune cells and stromal cells. The composition of the TME is actively orchestrated by the HRS cells via the production of different chemokines and interactions with the surrounding immune cells in the TME [111,127,128]. Many of the immune cells within the TME support the survival and proliferation of HRS cells, while others have the potential to eliminate the cHL tumor cells [129]. There is significant heterogeneity in the TME composition between individual cHL specimens, but T cells represent one of the more prevalent populations. These include mainly CD4^+^ T helper (Th) and regulatory T cells (Tregs), and to a lesser extent CD8^+^ cytotoxic T cells [127,129]. Single-cell expression profiling has revealed the expansion of Treg cell features in cHL as compared to reactive lymph nodes, displaying relatively high expression of LAG3^+^ and CTLA4^+^ Treg cell clusters in cHL samples [130]. Other non-Treg CD4^+^ T-cell clusters that are enriched in cHL samples include type 17 T helper (Th17) cells, which are more predominant in EBV-negative cases [130]. On the other hand, the proportion of type 1 T helper (Th1) cells is significantly enriched in lymphocyte-rich and mixed cellularity cHL [131]. Other quantitative approaches to investigate the cHL immune microenvironment involved mass cytometry, which showed concomitant expansion of Th1-polarized T effector cells and Tregs in cHL [132]. Due to the T cell-rich TME of cHL, minor T-cell clones can be detected with clonality testing in whole tumor tissue samples. These may represent clonal expansions of tumor- or EBV-reactive T cells.

A unique feature of cHL is the close proximity of rosetting CD4^+^ T cells. Such associations involve soluble factors and surface ligand–receptor interactions (e.g., CD40), and via pairs of adhesion molecules (i.e., CD54–CD11a and CD58–CD2). Furthermore, this CD40–CD40L interaction serves as an activation signal for the NF-κB pathway, which is important in cHL pathogenesis as described above. The rosetting T cells are also believed to protect the HRS cells from recognition and elimination by cytotoxic T cells and natural killer cells [127,133]. This is especially important in EBV-positive cHL, as such HRS cells express viral antigens that may be recognized by cytotoxic cells. However, HRS cells often have lost MHC class I and II molecules on their cell surface, which also contributes to immune escape via a reduced immunogenicity and impaired activation of T cells [129,134].

## 5. Conclusions and Future Perspectives

Although the molecular analysis of cHL has been hampered for many years by the low frequencies of malignant HRS cells, recent developments in sequencing technologies and analysis of cfDNA have made it possible to unravel the genetic features of cHL, which will further advance cHL diagnostics. This includes both improvements in clonality detection and the identification of recurrently mutated genes. In particular, the combination of IG and TR clonality testing may be of added value to correctly diagnose patients with cHL or T-non-Hodgkin lymphoma (T-NHL) with HRS-like cells. To further support the diagnosis of a possible T-NHL, targeted mutation analysis can be performed for recurrently mutated genes (e.g., *RHOA*, *TET2*, *DNMT3A* and *IDH2* for AITL [135]). With the advances of single cell analyses (whole exome and/or genome sequencing), additional insights in cHL driver mutations and clonal dynamics can be obtained.

Until now, no universal molecular marker has been identified that could serve as a disease-specific genetic marker for diagnosis and longitudinal monitoring of therapy response in cHL patients. One interesting development in molecular testing is the molecular profiling of ctDNA within the fraction of plasma cfDNA in lymphoma patients. Given the high concordance of mutations between cfDNA and tumor DNA, and higher variant allele frequencies in cfDNA, this represents a promising approach for future molecular studies. Especially for risk-stratification, monitoring of therapy response and early detection of recurrences, screening of robust biomarkers in routine diagnostics with targeted NGS or ddPCR will be highly valuable. The recurrently mutated *XPO1* gene might be a candidate biomarker, since the vast majority of *XPO1* mutations involve the E571K hotspot mutation. Detectable *XPO1* mutation by digital PCR at the end of treatment was shown to have potential predictive value for a shorter progression-free survival [65]. Additional patient-specific biomarkers can be retrieved from the identified IG gene rearrangements that could be monitored in cfDNA for the detection of MRD. Furthermore, cfDNA could be a highly interesting source to study the clonal evolution of relapsed disease, as previously described for different types of solid tumors [136,137,138,139] and B-cell lymphomas [140,141], including cHL [67]. Such longitudinal monitoring may provide information about the mechanisms of therapy resistance, and improve personalized treatment.

In conclusion, the major improvements in molecular testing of cHL have enhanced our understanding regarding cHL pathogenesis. These advances in NGS-based technologies will provide novel opportunities to improve diagnostics, disease monitoring and personalized therapies for cHL patients in daily clinical practice.

## Figures and Tables

**Figure 1 cancers-14-03222-f001:**
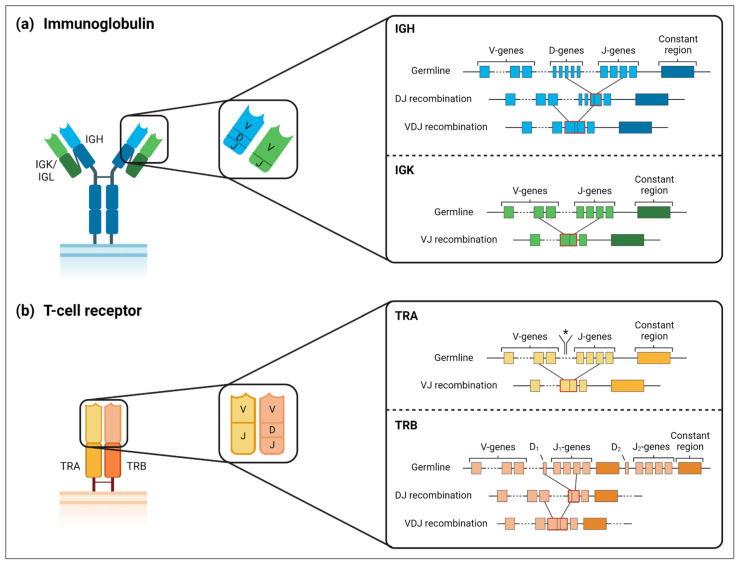
Schematic overview of immunoglobulin and T-cell receptor V(D)J gene rearrangements. (**a**) Immunoglobulin (IG) gene rearrangements generate a B-cell receptor, where IG heavy chain (IGH) genes are represented in blue, while IG kappa (IGK) and IG lambda (IGL) light chain genes are indicated in green. In this figure, only (potentially) productive IGH and IGK gene rearrangements are shown. In case of a non-productive IGK rearrangement, the kappa deleting element will rearrange with an upstream IGK-V gene or the Intron-RSS sequence (not shown in this figure). If both alleles result in a non-productive IGK light chain, the IGL light chain will rearrange, according to a similar process as IGK (not shown in this figure). Please note that the structure of the IGL chain is slightly different compared to the IGK chain shown here, since each IGL constant region is preceded by a single IGL-J gene. (**b**) V(D)J gene rearrangements of alpha/beta T-cell receptor. TR alpha (TRA) chain is represented in yellow, TR beta (TRB) chain in orange. Gamma/delta T-cell receptors are generated similarly, where TR gamma (TRG) chain undergoes VJ rearrangement similar to TRA, while TR delta (TRD) chain is generated by VDJ rearrangement, like TRB. Please note that the TRD gene cluster is located between the TRA-V and TRA-J gene clusters, as indicated with an asterisk. This TRD gene cluster will be deleted upon TRA gene rearrangements. For both IG and TCR, dark shades represent the constant regions, while the lighter shades correspond to variable regions, which are composed of rearranged V, (D) and J genes.

**Figure 2 cancers-14-03222-f002:**
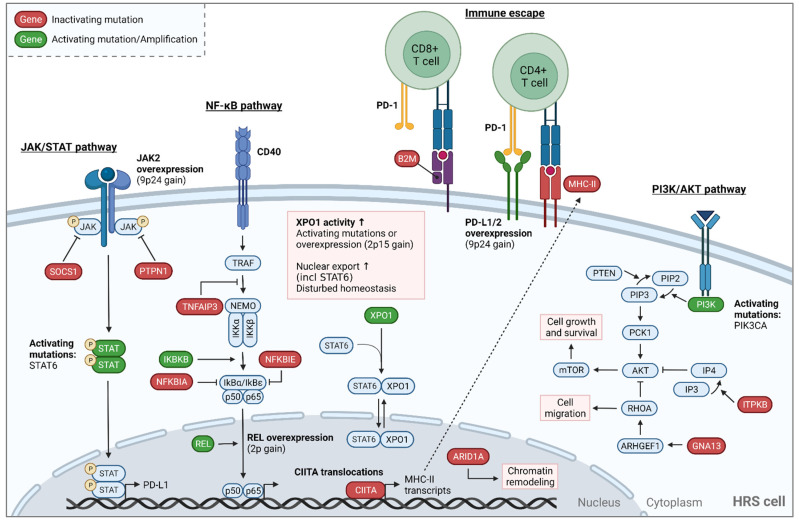
Schematic overview of the pathways affected by genetic alterations in classical Hodgkin lymphoma. The main affected pathways in the pathogenesis of cHL involve JAK/STAT, NF-κB, PI3K/AKT and immune evasion, caused by inactivating mutations (in red), activating mutations (in green), translocations and overexpression due to gene amplifications. Nuclear–cytoplasmic (e.g., XPO1) transport and epigenetic regulation (e.g., ARID1A) may be disturbed in HRS cells, contributing to oncogenesis.

**Figure 3 cancers-14-03222-f003:**
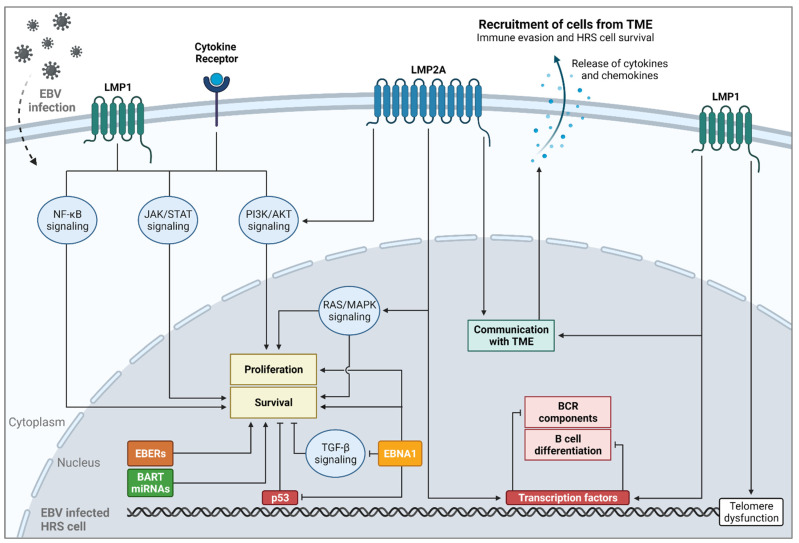
Schematic overview of EBV-associated gene products in the pathogenesis of classical Hodgkin lymphoma. EBV-associated transcripts and proteins are able to activate pathways and cellular processes that are recurrently affected by mutations in EBV-negative HRS cells. These alterations thereby play an important role in the pathogenesis and survival of EBV-positive HRS cells.

**Table 1 cancers-14-03222-t001:** Overview of clonality assessment in classical Hodgkin lymphoma by the analysis of IG and TR gene.

**IG Gene Rearrangements**					
	**Study Cohorts**		**Clone Detection (IG)**		
**Clonality Assay**	**Studies (n)**	**Samples (n)**	**Mean**	**Range**	**References**
Southern blot *	8	8–39	45%	0–87%	[19,20,21,22,23,24,25,26]
PCR-based *	3	32–212	30%	23–44%	[27,28,29]
BIOMED-2 *	8	12–58	57%	26–79%	[31,32,33,34,35,36,37,38,39]
NGS-based					
Tissue gDNA *	2	16–17	72%	56–88%	[50,51]
cfDNA	2	9–72	64%	38–89%	[50,52]
Enriched HRS cells ^#^	11	3–25	62%	0–100%	[17,18,26,42,43,44,45,46,47,48,49]
**TR Gene Rearrangements**					
	**Study cohorts**		**Clone detection (TR)**		
Clonality assay	Studies (n)	Samples (n)	Mean	Range	References
Southern blot *	8	8–39	15%	0–68%	[19,20,21,22,23,24,25,26]
BIOMED-2 *	1	58	17%	NA	[36]
Enriched HRS cells ^#^	4	3–19	28%	11–50%	[42,47,53,54]

* Whole tumor genomic DNA (gDNA). ^#^ Multiple assay types used. cfDNA: cell-free DNA.

**Table 2 cancers-14-03222-t002:** Overview of recurrent genetic aberrations in classical Hodgkin lymphoma and their effects.

Gene	Genetic Aberration(s) *	Pathways and Biological Processes
*PD-L1 and PD-L2*	CNAs (9p24 gain)	Immune evasion
*B2M, HLA-A/B*	Inactivating mutations	
*CIITA*	Translocations	
*JAK2*	CNAs (9p24 gain)	JAK/STAT signaling
*STAT6*	Activating mutations	
*SOCS1*	Inactivating mutations	
*CSFR2B*	Activating mutations	
*PTPN1*	Inactivating mutations	
*TNFAIP3*	Inactivating mutations and deletions	NF-κB pathway
*IKBKB*	Activating mutations	
*NFKBIA, NFKBIE*	Inactivating mutations	
*BIRC3*	CNAs (11q loss)	
*REL*	CNAs (2p gain)	
*GNA13*	Inactivating mutations	PI3K/AKT pathway
*ITPKB*	Inactivating mutations	
*RBM38*	Inactivating mutations	
*PIK3CA*	Activating mutations	
*XPO1*	Activating mutations	Nuclear–cytoplasmic transport
*TP53*	Inactivating mutations	Genomic stability
*ATM*	Inactivating mutations	
*ARID1A*	Inactivating mutations	Epigenetic regulation
*KMT2C*	Inactivating mutations	
*KMT2D*	Inactivating mutations	

* Activating mutations target oncogenes, while inactivating mutations target tumor suppressor genes.

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
