# Peer review of "Novel Approaches in Molecular Characterization of Classical Hodgkin Lymphoma"

_cancers, 2022, doi:10.3390/cancers14133222_

Round 1

Reviewer 1 Report

The authors provide a concise overview about the pathogenesis of classical Hodgkin lymphoma and of molecular approaches to study genetic aspects of the disease. This is a valuable addition to other recent reviews about the biology of Hodgkin lymphoma.

Minor points:

1) Fig. 1a: The stucture presented for the IGK/IGL light chain loci only fits to the IgK locus, but is incorrect for the IgL locus.  In the human IgL locus, each Clambda gene is preceded by its “own” Jlambda segment, i.e. the Jlambda segments are not in a row upstream of a single Clambda gene.

2) When discussing the various approaches to identify the clonal IgV gene rearrangements of HRS cells, it should be discussed that a potential caveat of the whole tissue section analysis is that a clonal rearrangement cannot unequivocally be assigned to the HRS cells. There is the possibility that there are other expanded B cell clones in the tissue, and/or that there is a biased amplification of V gene rearrangements in PCR studies with pools of cells. Thus, a (single cell) analysis of isolated HRS cells not only is more sensitive than a whole tissue analysis, but also more convincing and specific to identify the IgV genes of the HRS cells.

3) Table 1: The list of studies of isolated/enriched HRS cells for IgV gene rearrangements is not comprehensive. For example, a study by H. Kanzler et al (J Exp Med. 1996) is a landmark paper in this regard, and there is a study on lymphocyte-rich classical Hodgkin lymphoma that can be cited (A. Bräuninger et al., Cancer Res. 2003).

4) Lines 131-139: The first study describing the same HRS cell clone in different time points and biopsy sites of two Hodgkin lymphoma patients was by M. Vockerodt et al., Blood 1998.

5) Lines 161-169: Regarding the issue that Hodgkin lymphoma patients may harbor mutated clones from clonal hematopoiesis clonally independent from the HRS cell clone a recent study by A. Venanzi et al. (Blood Cancer Discovery, 2021) should be considered.

6) Lines 161-193: When discussing the studies on ctDNA in blood of Hodgkin lymphoma patients, it is recommended to also cite a recent quite large study by S. Sobesky et al. (Med, 2021).

7) Table 2: SOCS1 better fits to the JAK/STAT signaling gene group than to the Immune evasion gene group. 

Author Response

The authors provide a concise overview about the pathogenesis of classical Hodgkin lymphoma and of molecular approaches to study genetic aspects of the disease. This is a valuable addition to other recent reviews about the biology of Hodgkin lymphoma.

Author’s response: Thank you for the kind complement.

Minor points:

1) Fig. 1a: The stucture presented for the IGK/IGL light chain loci only fits to the IgK locus, but is incorrect for the IgL locus.  In the human IgL locus, each Clambda gene is preceded by its “own” Jlambda segment, i.e. the Jlambda segments are not in a row upstream of a single Clambda gene.

Author’s response: Thank you for the valuable remark, now only IGK rearrangements are shown in Figure 1, additional information about IGL locus is provided in the figure legend, and “IGK/IGL” in the box at the right side of the figure is adjusted to “IGK”.

2) When discussing the various approaches to identify the clonal IgV gene rearrangements of HRS cells, it should be discussed that a potential caveat of the whole tissue section analysis is that a clonal rearrangement cannot unequivocally be assigned to the HRS cells. There is the possibility that there are other expanded B cell clones in the tissue, and/or that there is a biased amplification of V gene rearrangements in PCR studies with pools of cells. Thus, a (single cell) analysis of isolated HRS cells not only is more sensitive than a whole tissue analysis, but also more convincing and specific to identify the IgV genes of the HRS cells.

Author’s response: This argument was intended by the statement that clonality testing on enriched and isolated single HRS cells increases the sensitivity of clonality detection. By adding also “specificity” we cover the issue that non-malignant B cell clones could be identified in whole tissue and IGV primers may show competition in pools of cells.

3) Table 1: The list of studies of isolated/enriched HRS cells for IgV gene rearrangements is not comprehensive. For example, a study by H. Kanzler et al (J Exp Med. 1996) is a landmark paper in this regard, and there is a study on lymphocyte-rich classical Hodgkin lymphoma that can be cited (A. Bräuninger et al., Cancer Res. 2003).

Author’s response: The studies of Kanzler et al (1996) and Bräuninger et al (2003) have been added to the table and text.

4) Lines 131-139: The first study describing the same HRS cell clone in different time points and biopsy sites of two Hodgkin lymphoma patients was by M. Vockerodt et al., Blood 1998.

Author’s response: The study of Vockerodt et al (1998) included only 2 patients, in the more recent publication of Obermann et al (2011) a larger cohort of 20 relapsed cHL patients was investigated. This is mentioned in the text more clearly now, without referring to the earlier study of Vockerodt et al, since the small numbers will provide no insight in the clonal relationship of recurrences as such.

5) Lines 161-169: Regarding the issue that Hodgkin lymphoma patients may harbor mutated clones from clonal hematopoiesis clonally independent from the HRS cell clone a recent study by A. Venanzi et al. (Blood Cancer Discovery, 2021) should be considered.

Author’s response:  This reference has been added.

6) Lines 161-193: When discussing the studies on ctDNA in blood of Hodgkin lymphoma patients, it is recommended to also cite a recent quite large study by S. Sobesky et al. (Med, 2021).

Author’s response: This reference has been added.

7) Table 2: SOCS1 better fits to the JAK/STAT signaling gene group than to the Immune evasion gene group.

Author’s response:  Indeed, SOCS1 has been moved to JAK/STAT signaling in Table 2.

Reviewer 2 Report

line 157:  Of note, ctDNA isolation from plasma after blood sampling requires special nucleic 157 acid preservation tubes (for example Roche or Streck cfDNA BCT) and processing of these 158 samples within 24 hours to avoid degradation of leukocytes and subsequent 159 contamination of the ctDNA fraction.

I believe at the cfDNA day a comparison of the different tubes was given, and while streck tubes were superior in preserving DNA, EDTA tubes functioned very well, as long they were processed within a certain time limit.

general comment: 

role of single cell whole genome sequencing? Is there/will there be a role?

beatiful images, clear, not too crowded, supporting the article. 

the immune micro-environment and its link to the molecular profiling/detection of HL is not completely clear to me. While interesting, what is the goal of its inclusion in the paper?

Author Response

Line 157:  Of note, ctDNA isolation from plasma after blood sampling requires special nucleic acid preservation tubes (for example Roche or Streck cfDNA BCT) and processing of these samples within 24 hours to avoid degradation of leukocytes and subsequent contamination of the ctDNA fraction.

I believe at the cfDNA day a comparison of the different tubes was given, and while streck tubes were superior in preserving DNA, EDTA tubes functioned very well, as long they were processed within a certain time limit.

Authors response:  Thank you for the valuable remark, the option of EDTA tubes has been added to the text as well, with the respective processing times.

general comment: role of single cell whole genome sequencing? Is there/will there be a role?

Author’s response:  A sentence has been added to the text about the potential role of single cell WES/WGS.

beatiful images, clear, not too crowded, supporting the article. 

Author’s response: Thank you for this compliment.

the immune micro-environment and its link to the molecular profiling/detection of HL is not completely clear to me. While interesting, what is the goal of its inclusion in the paper?

Author’s response: The goal is to provide some additional context that T cell clonality can be detected in cHL due to the T cell-rich TME, and we have added text to explain this better. In addition, there are the T-NHL cHL mimickers that may also yield T cell clones. The combination of IG- and TR-clonality testing (combined with mutation analysis) could be very informative for correctly diagnose patients with cHL or T-NHL.

Reviewer 3 Report

I would liekt o congratulate the authors on this overview of the different approaches to characterize the molecular composition of cHL. They authors described the different mutated genes and various techniques. The manuscript is well-written and illustrated.

Minor remarks:

1.Figure 1 legend:  IG heavy chain genes (IGH)...

2. Please mention the ongoing studies with NGS-based data collection.

3.Please discuss, how could we implement NGS analysis in our daily clinical practise.

Author Response

I would liekt o congratulate the authors on this overview of the different approaches to characterize the molecular composition of cHL. They authors described the different mutated genes and various techniques. The manuscript is well-written and illustrated.

Author’s response: Thank you for the kind compliment.

Minor remarks:

1.Figure 1 legend:  IG heavy chain genes (IGH)...

Author’s response: This has been adjusted in the figure legend.

  1. Please mention the ongoing studies with NGS-based data collection.

Author’s response: There are now many efforts in different research and clinical studies to collect NGS-based datasets in cHL, and we will not be able to provide an all-encompassing overview on these studies, since most of these studies are not publicly available.  A systematic review might be an option on publicly available datasources, but we feel that this would require a complete new manuscript, which falls outside the scope of this review article.

3.Please discuss, how could we implement NGS analysis in our daily clinical practise.

Author’s response: At present it is not yet recommended to perform (NGS-based) clonality testing directly at first presentation of cHL (unless indications for potential presence of T-NHL), as in most cases this disease is curable. However, at time of a second (and subsequent) presentation it could be of added value to determine the clonal relationship of the tumors (in case of unrelated maybe different treatment options?) and to check for potential misdiagnosis of AITL as being cHL. This can be done by performing additional TR(-NGS) clonality testing, combined with targeted mutation analysis and pathological revision of tumor tissue sections. In case a robust biomarker has been identified with the potential to improve (personalized) treatment of a primary tumor, the analysis of this biomarker in cfDNA can be used for risk stratification and for therapy response monitoring. We have eluted to these novel developments in our review, and especially in the section of future perspectives.